# Learned Half-Quadratic Splitting Network for MR Image Reconstruction

**Bingyu Xin**[1]                                                          bx64@rutgers.edu
**Timothy S. Phan**[2]                                Timothy.S.Phan@nyulangone.org
**Leon Axel**[2]                                               Leon.Axel@nyulangone.org
**Dimitris N. Metaxas**[1]                                               dnm@cs.rutgers.edu

[1] *Department of Computer Science, Rutgers University, Piscataway, NJ 08854, USA.*

[2] *Department of Radiology, New York University, New York, NY 10016, USA.*

## Abstract

Magnetic Resonance (MR) image reconstruction from highly undersampled $k$-space data is critical in accelerated MR imaging (MRI) techniques. In recent years, deep learning-based methods have shown great potential in this task. This paper proposes a learned half-quadratic splitting algorithm for MR image reconstruction and implements the algorithm in an unrolled deep learning network architecture. We compare the performance of our proposed method on a public cardiac MR dataset against DC-CNN, ISTANet$^+$ and LPDNet, and our method outperforms other methods in both quantitative results and qualitative results. Finally, we enlarge our model to achieve superior reconstruction quality, and the improvement is 1.00 dB and 1.76 dB over LPDNet in peak signal-to-noise ratio on $5\times$ and $10\times$ acceleration, respectively. Code for our method is publicly available at https://github.com/hellopipu/HQS-Net.

**Keywords:** MR reconstruction, $k$-space, Compressed sensing, Deep Learning, Cardiac

## 1. Introduction

Magnetic resonance imaging (MRI) has been widely used in clinical disease diagnosis as a non-invasive imaging technique with a high spatio-temporal signal-to-noise ratio (SNR). However, the main limitation for MRI is the slow acquisition procedure, which usually lasts between 15 to 90 minutes per subject. For dynamic cardiac MRI, subjects are required to hold their breath and stay still to reduce imaging artifacts during the acquisition process, which is challenging or even impossible for those with breathing difficulties.

In MRI physics, $k$-space is the 2D or 3D Fourier transform of the MR image, and MR raw data is acquired in $k$-space. The recent fast MRI techniques aim to reduce the MRI acquisition time by scanning undersampled $k$-space data, which are then used to reconstruct the MR images by applying an inverse Fourier transform. This data undersampling process violates the Nyquist Theorem, and therefore the reconstructed images will be heavily aliased, which will result in imaging artifacts and low SNR.

Traditional compressed sensing MR image reconstruction methods (Ma et al., 2008) (Ravishankar and Bresler, 2010) (**?**) (Lingala et al., 2011) are time-consuming, and the reconstruction quality is often not satisfactory. The recent use of deep learning methods for MR image reconstruction has resulted in improved reconstruction quality, higher SNR with significant efficiency gains at runtime.

In this work, we propose a deep learning method that is motivated by the half-quadratic splitting (HQS) algorithm (Geman and Yang, 1995) for compressed sensing MR image reconstruction. We train, validate, and test our approach on a publicly available cardiac MR dataset with a single-coil acquisition setting. We compare our method with three mainstream and high-performance methods (Schlemper et al., 2017; Adler and Öktem, 2018; Zhang and Ghanem, 2018) with acceleration factors of $5\times$ and $10\times$. The results demonstrate the improvements offered by our method in terms of the MR reconstruction image quality, the model size, and the efficient inference speed. We also provide a larger size model for improved image reconstruction quality.

## 2. Brief Literature Review

Over the past 15 years, compressed sensing (CS) methods for MR image reconstruction (Lustig et al., 2007) have been one of the most successful reconstruction methods by exploring the sparsity of the image using sparse transforms such as the wavelet transform. CS reconstructs MR images by iteratively increasing the sparsity in transform space and updating the denoised images. Based on CS, DLMRI (Ravishankar and Bresler, 2010) exploits adaptive patch-based dictionaries as a more sparse transform to improve the reconstruction performance. The Learned Iterative Shrinkage-Thresholding Algorithm (LISTA) (Gregor and LeCun, 2010) is a fast algorithm that approximates optimal sparse codes by replacing two pre-computed matrices in classical ISTA with learned ones.

Recently, the success of deep learning has further inspired research in MRI reconstruction (Wang et al., 2021). CNNs automatically extract features that have significantly better representational power compared to hand-crafted features used by conventional methods. In these deep CNN methods, deep unrolled networks has dominated in the MR reconstruction task. Schlemper et al. (2017) follows the iterative algorithm in DLMRI but replaces the dictionary learning reconstruction with a deep cascade of CNNs (DC-CNN). DC-CNN outperforms the conventional method significantly in both reconstruction quality and inference time and is a great baseline for today's $k$-space reconstruction research. ISTA-Net (Zhang and Ghanem, 2018) is proposed by mapping the traditional ISTA for optimizing a general $l_1$ norm CS reconstruction model into a deep network and can be essentially viewed as a significant extension of LISTA (Gregor and LeCun, 2010). LPDNet (Adler and Öktem, 2018) is also an iterative reconstruction scheme. It is inspired by Primal-Dual Hybrid Gradient (PDHG) algorithm (Sidky et al., 2012) where the primal and dual proximal operators are replaced by learned CNNs. LPDNet was originally proposed for tomographic data reconstruction, but it shows superior performance over other methods on recent MR reconstruction challenges (Muckley et al., 2021).

The PDHG algorithm used in LPDNet is a special case of the proximal gradient descent (PGD) method used in DC-CNN and ISTA-Net, while HQS is a more efficient algorithm when the forward model is linear and regularization is not smooth. While both PGD and HQS alternating between data consistency step and denoising step, HQS instead performs a full model inversion rather than a single gradient step for data consistency update (Kellman et al., 2020). Our proposed method with several model designs based on HQS shows superior reconstruction performance compared to all the models mentioned above.

## 3. Proposed Method

### 3.1. Problem Formulation

We want to to reconstruct the complex-valued MR image $x$ from single-coil undersampled measurements $y$ in $k$-space, such that:

$$y = F_u x + \epsilon \tag{1}$$

where $F_u = MF$, where $M$ is a cartesian undersampling mask in $k$-space, $F$ is the Fourier transform, $\epsilon$ is the acquisition noise. Eq.(1) is underdetermined, and hence the inversion is ill-posed. According to CS theory, we can estimate $x$ by formulating an optimization problem:

$$\min_x \frac{1}{2}\|y - F_u x\|_2^2 + \lambda R(x) \tag{2}$$

where $\|y - F_u x\|_2^2$ is the data fidelity term, $R(x)$ is a regularization term on $x$ and $\lambda$ is to adjust the regularization based on the noise level of $y$. For traditional CS-based methods, the regularization term $R(x)$ typically involves $l_0$ or $l_1$ norms in the sparse domain of $x$.

### 3.2. Half Quadratic Splitting (HQS) Algorithm

The variable splitting technique is usually adopted to decouple the fidelity term and regularization term (Geman and Yang, 1995) (Boyd et al., 2010). By introducing an auxiliary variable $z$, Eq.(2) is equivalent to the constrained optimization problem below:

$$\min_x \frac{1}{2}\|y - F_u x\|_2^2 + \lambda R(z), \quad \text{s.t.} \quad z = x \tag{3}$$

The HQS method(Geman and Yang, 1995) solves the following problem:

$$\min_{x,z} \frac{1}{2}\|y - F_u x\|_2^2 + \lambda R(z) + \frac{\mu}{2}\|z - x\|_2^2 \tag{4}$$

where $\mu$ is a penalty parameter. Eq.(4) can be solved in an iterative strategy, HQS optimizes $\{x, z\}$ in an alternating fashion by solving the following two subproblems separately:

$$x_{k+1} = \arg\min_x \|y - F_u x\|_2^2 + \mu\|x - z_k\|_2^2 \tag{5a}$$

$$z_{k+1} = \arg\min_z \frac{\mu}{2}\|z - x_{k+1}\|_2^2 + \lambda R(z) \tag{5b}$$

The fidelity term and regularization term are decoupled into Eq.(5a) and Eq.(5b), respectively. Eq.(5a) contains the fidelity term associated with a quadratic regularized least-squares problem, and the closed-form solution is given by:

$$x_{k+1} = (F_u^H F_u + \mu I)^{-1}(F_u^H y + \mu z_k) = z_k + \frac{1}{1+\mu}F_u^H(y - F_u z_k) \tag{6}$$

where $F_u^H$ is the Hermitian of $F_u$, $I$ is the identity matrix. Eq.(5a) can be easily solved by Eq.(6). However, solving Eq.(5b) efficiently and effectively is non-trivial.

### 3.3. Learned HQS

Our goal is to derive a learned reconstruction scheme inspired by HQS. Motivated by former works (Sun et al., 2016; Schlemper et al., 2017; Adler and Öktem, 2018), our approach replaces Eq.(5b) by a parametrized CNN learned from training data. Inspirit of deep residual learning (He et al., 2016), our CNN updates $z_{k+1}$ from $z_k$, so Eq.(5b) can be written as below:

$$z_{k+1} = z_k + \text{CNN}(z_k, x_{k+1}) \tag{7}$$

We also use a buffer design for $z$ adapted from Adler and Öktem (2018). The buffer verion of $z$ is denoted as $f$, which is initialized as $f_0 = [x_0, ..., x_0]_m$, where $m$ is the size of buffer, $x_0$ is the zero-filled image. Only the first data in the buffer $f_k$, which is denoted as $f_k^{(0)}$ is used to update $x_{k+1}$ in Eq.(6), the other data in the buffer is used as additional information for updating $f_{k+1}$ using Eq.(7). The buffer design (additional storage) is originally used in quasi-Newton methods to accelerate convergence (Liu and Nocedal, 1989). Our proposed method is outlined in Algorithm 1.

Figure 1 shows an overview of our proposed unrolled network architecture to solve the MR reconstruction problem. The input to the network is the buffer data $f_0$, which is the concatenation of $m$ copies of the complex-valued zero-filled image $x_0$, the channel size of $f_0$ is $2m$. The proposed network consists of $n$ reconstruction blocks, which correspond to $n$ iterations in the HQS algorithm. In each reconstruction block, the first data $f_{i-1}^{(0)}$ in $f_{i-1}$ is used for updating Eq.(5a) by the solution given in Eq.(6) denoted as an update operation in the figure. The updated $x_i$ will be concatenated with $f_{i-1}$ as the input for updating Eq.(5b), while the updating module for Eq.(5b) is a learnable CNN with 6 convolutional layers denoted as red and green arrows shown in the block. Deep residual learning (He et al., 2016) is adapted in the block for better learning performance. The output of the network $f_n^{(0)}$ is the final reconstructed MR image by our method.

Although we discuss Algorithm 1 in the context of single-coil cartesian MRI, it can also be extended to multi-coil non-Cartesian MRI with minor modifications. When $k$-space is sampled by non-Cartesian trajectories, we need to replace Fourier transform operator by Non-Uniform Fourier transform operator; While in the multi-coil setting, $(F_u^H F_u + \mu I)$ in Eq.(6) is not analytically invertible, in this case, we need to use conjugate gradient optimization to solve Eq. (5a) (Aggarwal et al., 2018).

---

**Algorithm 1:** Learned Half-Quadratic Splitting

---

**Input:** zero-filled MR image $x_0$, Fourier operator $F_u, F_u^H$, iterations $n$, buffer size $m$
**Output:** reconstructed image $f_n^{(0)}$
Initialize $f_0 = [x_0, ..., x_0]_m$, $\mu$ and $\Gamma_{\theta_i}$ are learned parameters
**for** $i = 1$ **to** $n$ **do**
  $\quad x_i \leftarrow f_{i-1}^{(0)} + \frac{1}{1+\mu} F_u^H (y - F_u f_{i-1}^{(0)})$;
  $\quad f_i \leftarrow f_{i-1} + \Gamma_{\theta_i}(f_{i-1}, x_i)$;
**end**

---

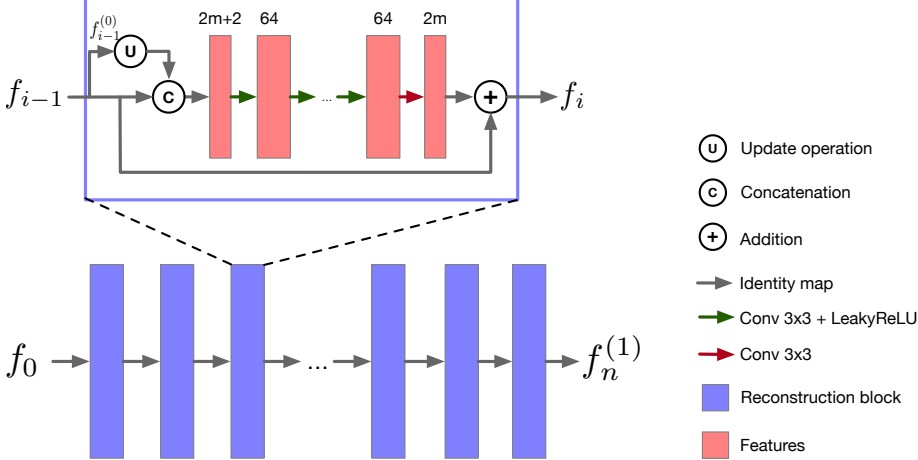

Figure 1: Overview of the proposed reconstruction network architecture. The input data $f_0$ is the concatenation of $m$ copies of the complex-valued zero-filled images $x_0$, the network contains $n$ reconstruction blocks, and the final reconstructed image is $f_n^{(1)}$. In each block, the update operation is utilized by Eq.(6), a learnable CNN with six convolutional layers denoted as red and green arrows is used for updating Eq.(5b). The numbers in the figure denote the channels of the output features of the CNN layers. Skip connection is adapted for better learning performance.

### 3.4. Loss function

Following Pezzotti et al. (2020), we deploy a compound loss of MS-SSIM (Wang et al., 2003) loss and L1 loss while training our model:

$$\mathcal{L} = \gamma \mathcal{L}^{\text{MS-SSIM}}(x_{rec}, x_{gt}) + (1 - \gamma)\|x_{rec} - x_{gt}\|_1 \tag{8}$$

where $\mathcal{L}^{\text{MS-SSIM}}$ is MS-SSIM loss, $x_{rec}$ is the reconstructed image, $x_{gt}$ is the ground truth image, $\gamma$ is the weight for MS-SSIM loss.

## 4. Experiments

### 4.1. Dataset

We use an open-access complex-valued Cardiovascular MR (OCMR) dataset[1] (Chen et al., 2020) in our experiments. The dataset provides multi-coil $k$-space data from 74 fully sampled cardiac cine series. In this paper, we only experiment on 2D image slices with a single-coil setting. We first generate the emulated single-coil $k$-space data from the OCMR dataset using the method described in Tygert and Zbontar (2020). Each 2D image slice is processed to the size of [2,192,160], where the first dimension stores the real part and the imaginary part of the complex value. The train, validation, and test sets have 1874, 544, and 1104 slices, respectively, from a disjoint set of subjects.

---

1. https://ocmr.info/

## 4.2. Metrics

In this work, we use three commonly used metrics in the MR reconstruction task: normalized root-mean-square error (NRMSE), peak signal-to-noise ratio (PSNR), and the structural similarity index measure (SSIM) to evaluate the reconstruction quality of different methods. These metrics are computed against the fully sampled MR images. A good reconstruction will have low NRMSE, high PSNR and SSIM values.

## 4.3. Experiment settings

We implemented two versions of our method in the experiments; one model is the same as shown in Figure 1, denoted as *HQSNet*, while the other model is much larger, which replaces the regular CNN with a modified U-net (Zhang et al., 2021) that is shared between different reconstruction blocks, denoted as *HQSNet-Unet*. We use this model to get the best reconstruction quality without considering the model size. We compare our method against three published methods, soft format DC-CNN (Schlemper et al., 2017), ISTANet[+] (Zhang and Ghanem, 2018) and LPDNet (Adler and Öktem, 2018).

For fair comparison between HQSNet and three other models, the number of iterations, the number of convolution layers in each reconstruction block and the number of output channel in intermediate convolution layers are all set to 8, 6 and 64, respectively. The buffer size $m$ is set to 5. We use Adam optimizer, compound loss ($\gamma = 0.84$), and the learning rate is set to 0.001 for all models. All other hyper-parameters may influence the fairness are all set the same. All models are trained from scratch for two different acceleration factors of 5× and 10×. We use random cartesian sampling masks throughout training and a fixed cartesian mask when validating and testing. Input data is the zero-filled image normalized such that the magnitude of 99th percentile pixel in the image is equal to 1, and we apply random crop and random affine transformation as data augmentation. More details about experiments can be found in our public code.

## 5. Results and Discussion

**Quantitative Results** Quantitative results of different models are summarized in Table 1. As a widely used baseline model in MR reconstruction, the reconstructed images using DC-CNN far surpass the zero-filled images of the NRMSE, PSNR, and SSIM values. ISTANet[+] is slightly better than DC-CNN. LPDNet achieves better results than ISTANet[+] by updating reconstruction in both image and $k$-space domain. Our proposed HQSNet outperforms these methods in all metrics, which shows its effectiveness, and the HQSNet-Unet further improves the reconstruction quality owing to its larger model capacity.

**Qualitative Results** Figure 2 shows reconstructed MR image samples in the test set of different models on acceleration factors of 5× and 10×. On 5× acceleration, the zero-filled image is corrupted aliasing artifacts. The reconstructed images by different models are much better than the zero-filled image and visually similar to the ground truth. However, we can still find subtle differences using the zoomed area and error maps. Our proposed two models get slightly better quality than the other three methods; on 10× acceleration, the aliasing artifact in the zero-filled image becomes more prominent. DC-CNN and LPDNet only generate unsatisfied recoveries, HQSNet is slightly better, and HQSNet-Unet is much

Table 1: Mean/std results of different methods on two acceleration factors. The best and second best results are highlighted in red and blue colors

| Acc | Metric | Zero-Filled | DC-CNN | ISTANet$^+$ | LPDNet | HQSNet | HQSNet-Unet |
|---|---|---|---|---|---|---|---|
| | NRMSE(%) | 41.49/4.29 | 15.99/3.10 | 16.05/2.99 | 15.77/2.83 | 14.98/2.84 | 14.10/2.71 |
| 5× | PSNR(dB) | 25.20/1.63 | 33.61/2.98 | 33.56/2.95 | 33.70/2.90 | 34.17/2.96 | 34.70/3.06 |
| | SSIM | 0.603/0.049 | 0.875/0.045 | 0.884/0.040 | 0.887/0.039 | 0.895/0.038 | 0.904/0.039 |
| | NRMSE(%) | 59.50/4.72 | 28.70/4.11 | 28.26/3.78 | 27.99/4.13 | 26.19/3.95 | 22.99/4.09 |
| 10× | PSNR(dB) | 22.05/1.66 | 28.45/2.67 | 28.57/2.59 | 28.67/2.87 | 29.25/2.77 | 30.43/3.02 |
| | SSIM | 0.469/0.063 | 0.731/0.065 | 0.752/0.061 | 0.758/0.070 | 0.776/0.064 | 0.804/0.069 |

Table 2: Comparison of FLOPs and number of parameters for different models. The best and second best results are highlighted in red and blue colors

| Models | DC-CNN | ISTANet$^+$ | LPDNet | HQSNet | HQSNet-Unet |
|---|---|---|---|---|---|
| FLOPs (G) | 36.81 | 36.81 | 78.99 | 39.35 | 660.72 |
| # of param (M) | 1.20 | 1.20 | 2.58 | 1.28 | 32.65 |

better, especially in the zoomed area where the reconstructed myocardium wall and the boundary are more precise and closer to the ground truth.

**Model Comparison**  Table 2 gives a summary of the number of model parameters and inference speed of these methods. Compared to DC-CNN and ISTANet$^+$, HQSNet can achieve better results with small additional parameters and FLOPs. The HQSNet-Unet with a larger capacity can reconstruct images of much higher quality at the cost of model size and inference time. It's worth noting that we can easily balance between the model size and reconstruction quality by choosing an appropriate buffer size $m$, the number of iterations, the number of convolution layers in each reconstruction block, and the convolution channel size. We can further improve the reconstruction quality of the HQSNet-Unet by increasing these hyper-parameters or replace Unet with more powerful models.

**Ablation Study**  For ablation study, we address three main differences between our proposed HQSNet and DC-CNN. The first is the order of data consistency step (DC) and denoiser step (DN) in each iteration, DC-CNN is DN first while HQSNet is DC first. Theoretically it should make no difference to the result if the algorithm converges, but in our practice, DC first is slightly better than DN first; The second difference is how we implement DN step, DN in HQSNet is implemented by Eq.(7), while DC-CNN is implemented by $z_{k+1} = x_{k+1} + \text{CNN}(x_{k+1})$, we find our DN design is more effective because it updates $z_{k+1}$ from $z_k$ and CNN only needs to represent to residual of $z$. The third is the buffer design for $z$, which is discussed in model description. Table 3 shows the effect of each design in HQSNet.

## 6. Conclusion

This paper proposed a learned HQS method for MR image reconstruction. Our method outperforms other reconstruction methods with higher reconstruction quality, fewer model parameters, and faster speed. We also provide a more extensive version model to achieve

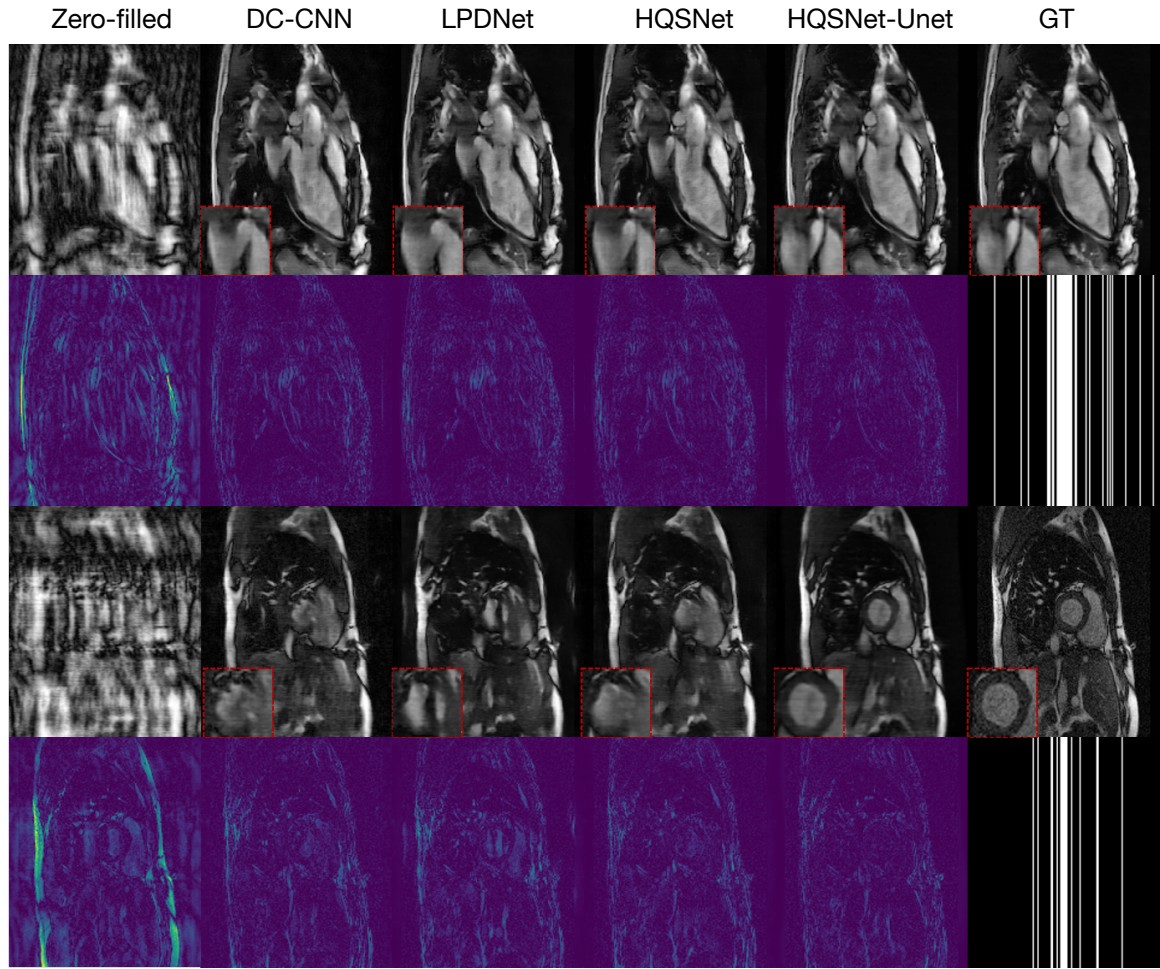

Figure 2: Reconstruction samples. The first two rows and the last two rows show the reconstructed images and error maps on the acceleration factors of $5\times$ and $10\times$, respectively. The first five columns show the reconstruction and corresponding error maps of different models, from left to right is zero-filled, DC-CNN, LPDNet, our proposed HQSNet model, and HQSNet-Unet model. The red square on the bottom left in each reconstructed image shows the zoomed area. The last column shows the ground truth (GT) images and the cartesian sampling masks.

Table 3: Ablation study on HQSNet design

| Model | DC_first | DN design | no buffer | buffer=3 | buffer=5 | buffer=7 | PSNR/SSIM |
|---|---|---|---|---|---|---|---|
| DC-CNN | | | ✓ | | | | 28.45/0.731 |
| - | ✓ | | ✓ | | | | 28.48/0.740 |
| - | ✓ | | | | ✓ | | 29.00/0.767 |
| - | ✓ | ✓ | ✓ | | | | 29.00/0.756 |
| | ✓ | ✓ | | ✓ | | | 29.07/0.767 |
| HQSNet | ✓ | ✓ | | | ✓ | | **29.25/0.776** |
| | ✓ | ✓ | | | | ✓ | 29.25/0.774 |

visually more pleasing reconstruction results. We validate the effectiveness of our method on a public cardiac MR dataset. In future research, we will extend our approach to dynamic cardiac MR data acquired with multi-coil and radial sampling masks, which is a more realistic scenario.

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
