# OpenReview forum: "Learned Half-Quadratic Splitting Network for MR Image Reconstruction"
_MIDL.io/2022/Conference — MIDL 2022_

### Official Review · Reviewer_aere · 2022-01-18

**Confidence:** 5
**Preliminary Rating:** 3

**Summary:**

The authors propose Half-Quadratic Splitting (HQS) network for supervised neural network training of CS-MRI reconstruction. The method is inspired by the conventional and widely used HQS algorithm, where the authors replace the proximal gradient operator with respect to the regularization term with parameterized neural network. The results show promising progress, and it seems to me that the method is sound. However, the paper would much benefit from discussing the relevant connections and motivations for replacing the proximal gradient step. Unless the motivation is clarified, the paper can be easily cast as just another method which replaces one of the CS routines with a neural network (e.g. ADMM-net, ISTA-net, etc.).

**Strengths:**

1. The paper is relatively easy to understand and follow.

2. I believe the paper is the first to implement HQS via unrolled iterations of neural network forward pass. The implementation seems sound.

3. Results are promising.

**Weaknesses:**

1. Lack of motivation. Since the advent of ADMM-nets, there have been many works which replace the proximal gradient steps (e.g. ADMM-net, ISTA-net, etc.) or the denoising steps (e.g. PnP, RED) via neural networks. All show promising results, and there are relatively many works which are within the context of CS-MRI. Compared to the previous works, how does the proposed method differ other than that they used HQS? Why would this be beneficial?

2. Method does not contain enough details. For example, the authors mention using a *buffer*, which is not sufficiently explained. I would advise elaborating on the algorithm in order to make the paper self-contained.

3. Adding on 2, I would also suggest adding a discussion on the actual proximal gradient descent, and taking a forward pass via $\Gamma_\theta$. Maybe some intuitions or visualizations of what kind of regularization $\Gamma_\theta$ imposes would be interesting.

4. The authors do not compare against methods that are the most similar to their approach. It would be much more convincing if the authors show that the proposed HQS-net is indeed better than or competitive to prior methods such as ADMM-net or ISTA-net.

**Deanonymize Review:**

no

**Final Rating After The Rebuttal:**

3: Borderline

**Justification Of The Final Rating:**

My main concern with the paper was with the lack of motivation and intuition. Reading the manuscript in its current form, it is still very hard to find **why** unrolling the HQS algorithm would be beneficial over other unrolling-based algorithms. At the least, I expected some conjectures and intuitive explanations. Unfortunately, while the authors performed extensive ablation studies to show the superiority of their design choices, these seem to be engineering prowess that could be applied to other existing methods to improve their performance. I keep my score as 3.

**Paper Type:**

methodological development

**Questions To Address In The Rebuttal:**

Clarifying the motivation and the connection between the proximal gradient step and the proposed neural network update step would be crucial. Other major questions are listed in the weaknesses session.

**Special Issue:**

no

---

### Official Review · Reviewer_p9qc · 2022-01-21

**Confidence:** 4
**Preliminary Rating:** 5
**Recommendation:** Poster

**Summary:**

The paper proposes a half-quadratic splitting network for Magnetic Resonance (MR) image reconstruction. They compared their method against two other deep-learning-based MR reconstruction methods (DC-CNN and LPDNet) using a public cardiac MR dataset. They experimented with their method for  MR acceleration factors of 5 and 10. They achieved the best quantitative results (SSIM, NRMSE, PSNR) while having the fewest number of learnable parameters and the second-fastest run time.

**Strengths:**

The paper is well-written and easy to follow. The experimental setup is sound. The solution proposed has a solid mathematical basis. It basically replaces one of the subproblems of the half-quadratic splitting formulation by a neural network.

**Weaknesses:**

- The two methods compared are relatively older in the field of deep-learning-based MR image reconstruction (2017 and 2018)
- The authors used a public dataset, but there are public benchmarks (fastMRI and Calgary-Campinas dataset), which would have been more appropriate to put the work in context.
- In the method explanation, it wasn't clear to me why the input is the concatenation of copies of the complex-valued zero-filled image.

**Deanonymize Review:**

no

**Detailed Comments:**

- The references should be standardized (MR shows in lowercase multiple times, etc.)

- Metrics and processing times should report not just the mean value, but also the standard-deviation

**Paper Type:**

methodological development

**Questions To Address In The Rebuttal:**

- Why the authors did not report the standard deviation for the results?

- Please explain why the input to the proposed model is the concatenation of copies of the complex-valued zero-filled image

- Please justify the use of the dataset chosen in the experiments.

**Special Issue:**

yes

---

### Official Review · Reviewer_fcSy · 2022-01-24

**Confidence:** 3
**Preliminary Rating:** 2
**Recommendation:** Poster

**Summary:**

This paper proposes a deep learning MRI reconstruction method based on half-quadratic splitting. Each unrolled iteration in the network first applies a closed form update based on the data fidelity portion of the HQS optimization and then applies convolutions to approximate the regularization update. The network is trained such that the output of several repetitions of this procedure is close to a ground truth image based on MS-SSIM and L1 distance. The method is demonstrated on a dataset of cardiac MRI scans, where it outperforms two previously proposed deep learning based methods.

**Strengths:**

- **The exposition of the half quadratic splitting algorithm is well-written.** Section 3.2 in particular is very clear and easy to follow.
- **The proposed strategy does seem to outperform the two baselines implemented.** Both the numerical results in Table 1 and the visual figures in Fig. 2 show reasonably clear improvements over the baselines.

**Weaknesses:**

- **The proposed network architecture does not exactly map onto the formulation derived in the paper.** Update equation 5(b) computes an estimate of $z_{k+1}$ based only on $x_{k+1}$. However, in algorithm 1, the update of $f_i$ is computed based on the previous $f_{i-1}$ in addition to $x_i$. So, to me it is not clear that the development in equation 5 applies exactly to the learning procedure in this paper.
- **The proposed network architecture is largely the same as that explored in previous work.** In particular, the formulation in this paper looks very similar to me as that in MoDL, cited in this paper. Equations 10(a-b) in the MoDL paper correspond to equation 5(a-b) in this paper, and both papers use a network to approximate the updates corresponding to the (b) equations while manually applying the (a) equations. As far as I can tell, there are three main differences between this work and MoDL.:
    1. In this paper, the update of the auxiliary variable is based on both the solution to the (a) equation and on the previous value of the auxiliary variable, while in the MoDL paper, the update of the auxiliary variable is based on the solution to the (a) equation alone.
    2. In this paper, the loss is a combination of MS-SSIM and L1 loss, while in MoDL, the loss is a pure L2 loss.
    3. In MoDL, the weights are shared across all iterations.

    Thus, in order to understand the contribution of this paper, we would need to understand how the proposed HQS scheme does in comparison to MoDL on the same dataset, and we need to understand which of the above three changes are driving any difference in performance.
- **Not enough ablation tests are performed to understand the improvement of this method over LPD-Net.** Though I am less familiar with LPD-Net, from a read of the paper, it seems that the major differences with this work are:
    1. The papers rely on two different optimization schemes (HQS in this paper vs. the primal-dual scheme in that paper).
    2. The papers train via two different loss functions (MS-SSIM and L1 in this paper and L2 in that paper).

    To understand whether the proposed HQS scheme is beneficial, we would need to see a comparison which controls for the different loss terms used to train the networks.

**Deanonymize Review:**

no

**Detailed Comments:**

- The literature review currently reads as a long list of methods. It would really help the reader to group these methods by strategy (e.g. all unrolled optimization-based methods listed together) along with an explanation which places the proposed HQS scheme in context — it should identify that HQS is another unrolled method and give the reader some intuition for why we expect the HQS approach to improve on other unrolled optimization-based methods.
- I believe Algorithm 1 is currently written incorrectly. First, $z_i-1$ is used in the update equation for $x_i$, but $z$ variables are not used in the second update equation. I believe the $f$s in the second update equation should be $z$s (if not, I didn’t under the algorithm, and this may need more explaining). Also, this algorithm should specify the initialization strategy for $z_0$.
- Please provide standard deviations in Table 1.

**Final Rating After The Rebuttal:**

4: Weak Accept

**Justification Of The Final Rating:**

The authors improved the description and clarity of the proposed algorithm and incorporated an ablation study which alleviated most of my initial concerns about this paper. As mentioned in my response below, I think the technical results, especially with the new ablation studies, do show and explain an improvement over DC-CNN, which is a widely-used baseline for the MR reconstruction problem. However, there is still significant room to improve this paper with clearer writing (more explicitly explaining the key differences between the proposed method and previous approaches in the manuscript text) and discussion of any intuition for why we see an improvement over LPD-net, which is the best performing baseline.

**Paper Type:**

methodological development

**Questions To Address In The Rebuttal:**

- **Have I missed or misunderstood any differences between the MoDL formulation and that in this paper?** If there are substantial differences that I have missed or if there is a reason that MoDL and this method are not comparable on this dataset, this may affect my rating.
- **What is the rationale for updating $f_i$ based on $f_{i-1}$ in addition to $x_i$**? If I have missed a well-justified reason for doing this, I would be more convinced that the network is an implementation of the described HQS scheme.

**Special Issue:**

no

---

### Meta-Review · Area_Chair_3xj6 · 2022-02-20

**Recommendation:** Accept (Poster)
**Confidence:** 3

**Metareview:**

The authors introduce a deep learning-based MR image reconstruction method with a trainable half-quadratic splitting (HPS) algorithm.
In particular, the method replaces the proximal gradient operator with respect to the regularization term with a parameterized neural network.

The authors demonstrate its benefits empirically both in speed and reconstruction quality with respect to relevant baselines and competing methods. All reviewers agree that the proposed method is well-motivated, and the empirical improvements are significant. Although the initial reviews consistently showed some concerns over the lack of understanding of where the improvement over the baselines comes from, the authors have addressed by adding a comprehensive ablation study to the updated manuscript in the rebuttal phase.

One outstanding criticism from Reviewer 3, which I believe is very reasonable, is the lack of discussions on why unrolling the HQS algorithm would be beneficial over other unrolling-based algorithms. Given the strong empirical results, I am leaning towards recommending acceptance, but I strongly encourage the authors to provide some conjectures/explanations for the superiority of their method.

---

### Decision · Program_Chairs · 2022-02-28

Accept